# Atomic-scale observation of structural and electronic orders in the layered compound α-RuCl₃

M. Ziatdinov[1,2], A. Banerjee[3], A. Maksov[1,4], T. Berlijn[1,5], W. Zhou[6], H.B. Cao[3], J.-Q. Yan[6,7], C.A. Bridges[8], D.G. Mandrus[6,7], S.E. Nagler[3,4], A.P. Baddorf[1,2] & S.V. Kalinin[1,2,4]

A pseudospin-1/2 Mott phase on a honeycomb lattice is proposed to host the celebrated two-dimensional Kitaev model which has an elusive quantum spin liquid ground state, and fascinating physics relevant to the development of future templates towards topological quantum bits. Here we report a comprehensive, atomically resolved real-space study by scanning transmission electron and scanning tunnelling microscopies on a novel layered material displaying Kitaev physics, α-RuCl₃. Our local crystallography analysis reveals considerable variations in the geometry of the ligand sublattice in thin films of α-RuCl₃ that opens a way to realization of a spatially inhomogeneous magnetic ground state at the nanometre length scale. Using scanning tunnelling techniques, we observe the electronic energy gap of $\approx 0.25\,eV$ and intra-unit cell symmetry breaking of charge distribution in individual α-RuCl₃ surface layer. The corresponding charge-ordered pattern has a fine structure associated with two different types of charge disproportionation at Cl-terminated surface.

[1] Center for Nanophase Materials Sciences, Oak Ridge National Laboratory, Oak Ridge, Tennessee 37831, USA. [2] Institute for Functional Imaging of Materials, Oak Ridge National Laboratory, Oak Ridge, Tennessee 37831, USA. [3] Quantum Condensed Matter Division, Oak Ridge National Laboratory, Oak Ridge, Tennessee 37831, USA. [4] Bredesen Center for Interdisciplinary Research, University of Tennessee, Knoxville, Tennessee 37996, USA. [5] Computer Science and Mathematics Division, Oak Ridge National Laboratory, Oak Ridge, Tennessee 37831, USA. [6] Material Science & Technology Division, Oak Ridge National Laboratory, Oak Ridge, Tennessee 37831, USA. [7] Department of Material Science and Engineering, University of Tennessee, Knoxville, Tennessee 37996, USA. [8] Chemical Sciences Division, Oak Ridge National Laboratory, Oak Ridge, Tennessee 37831, USA. Correspondence and requests for materials should be addressed to M.Z. (email: ziatdinovma@ornl.gov) or to A.B. (email: banerjeea@ornl.gov) or to S.V.K. (email: sergei2@ornl.gov).

The peculiar interplay between spin-orbit effects, electron–electron interaction, and subtle lattice distortions is expected to produce an elaborate phase diagram for the $4d^5$ transition metal layered compound $\alpha$-RuCl$_3$, which to date remains inadequately understood[1–4]. The proposed state of an electron on the honeycomb lattice of edge-shared RuCl$_6$ octahedra is described by a spin-orbit entangled $J_{eff} = 1/2$ Mott state[1,4,5] that can potentially host novel exotic quantum phases, such as a room temperature Quantum Spin Hall Effect[6] and Kitaev quantum spin liquid (QSL) behaviour[3,4,7,8]. The elementary excitations associated with the Kitaev QSL model include Majorana fermions[9–12], which are relevant in the context of a topological quantum computer[13]. Recent identification of a magnetic phase proximate to the Kitaev QSL in neutron scattering experiments[3] have further fueled an excitement over a breakdown of the putative classical physics in the $\alpha$-RuCl$_3$ system.

As the nearest-neighbour coupling between $J_{eff} = 1/2$ moments in $4d^5$ and $5d^5$ honeycomb compounds is highly sensitive to a small distortion of the 90° metal-ligand-metal bond[14–16], a rigorous understanding of the local structural properties of the $\alpha$-RuCl$_3$ compound is required for exploring a potential departure from the Kitaev QSL phase and realization of a spatially inhomogeneous magnetic ground state in the presence of lattice disorder. An equally important factor in establishing the atomic scale phenomenology for the $\alpha$-RuCl$_3$ system is a study of its local electronic behaviour. Whereas most of the experiments have so far focused on bulk magnetic properties of $\alpha$-RuCl$_3$, the experimental research on its nanoscale electronic properties, including a potential role of unit-cell scale charge (re)distribution in determining the final ground state and the associated exotic phenomena in the Kitaev limit, is scarce. In addition, exploring properties of thin films, the surface effects in $\alpha$-RuCl$_3$, and their relationship to the bulk parent compound can provide important clues for potential applications of 'Kitaev materials' in next generation of 2D nanoscale quantum electronic devices for the post-silicon era.

While, most of the studies in the material have concentrated on reciprocal space and bulk properties[2,3,17–19], to explore the many complexities of this system, it is important to be able to visualize directly its local structural and electronic orders in real space.

In the following, we employ a combination of *in-situ* scanning transmission electron microscopy (STEM) and *in-situ* scanning tunnelling microscopy (STM) on exfoliated/cleaved $\alpha$-RuCl$_3$ samples as ideal tools for detailed evaluation of structural and electronic parameters of the system with a sub-nanometre precision. To date the only existing in literature atomically resolved STM study on $\alpha$-RuCl$_3$ reported an apparently strong lattice distortions on the surface of the sample[20]. However, as those STM measurements were performed in the ambient environment and the samples were not cleaved *in situ*, it remains unclear whether the observed structure peculiarities were the intrinsic properties of the material or caused by some extrinsic effects, such of sample and/or STM tip contamination. The STM and STEM measurements performed here are further supported by neutron scattering, first principles calculations, and a multivariate statistical analysis. This combination of both reciprocal space and real space tools, together with a computational effort, provides an excellent starting point in establishing the structure-property relationship in $\alpha$-RuCl$_3$ system. In this study, we find that the exfoliated thin films (thickness $\approx 15 \sim 30$ layers) can be characterized by $P3$-type space group, which is explained with a help of neutron diffraction data as due to transition from $C2/m$ to $P3_1$ type stacking order at above $\sim 150$ K. For each individual $\alpha$-RuCl$_3$ layer we found that in addition to a uniform octahedral distortion reported earlier for

bulk $\alpha$-RuCl$_3$ crystals, there is a persistent local inhomogeneity in the ligand geometries. Such nanometre scale perturbations in structural order, inferred from the STEM data with a subpixel precision and not typically accounted for by theory, can have an important role for determining collective spin-orbital state of the system in Kitaev-Heisenberg model. The STM spectroscopic measurements in the paramagnetic phase of the *in-situ* cleaved $\alpha$-RuCl$_3$ samples demonstrated a presence of $\approx 0.25$ eV gap in density of states of the individual surface layer, which is explained by a realization of $J_{eff} = 1/2$ Mott state. Furthermore, our STM probe into the atomic-scale surface electronic structure revealed a striking anisotropy in a charge distribution along the Ru–Cl–Ru hopping paths, indicating that the bond-dependent behaviour in this system may appear at the temperatures considerably higher than the estimated strength of the Kitaev coupling. In addition, we observed a fine structure of the corresponding charge ordered state characterized by a coexistence of trigonal-like and dimer-like patterns of charge disproportionation at the Cl-terminated surface.

## Results

**Structure (neutron and STEM).** We start with a real-space characterization of $\alpha$-RuCl$_3$ films using high-resolution annular dark field STEM measurements. A representative Z-contrast STEM image (Z is atomic number) of $\alpha$-RuCl$_3$ film (thickness $\approx 30$ layers) taken perpendicular to the sample's $c^*$-axis at $T = 295$ K is shown in Fig. 1a. The image consists of a periodic repetition of an effective unit cell containing six brighter columns and three less bright columns. The registration of higher and lower Z atom columns with the $P3_112$ structural model for $\alpha$-RuCl$_3$ proposed in ref. 3 provides a good initial match to the entire image (see Supplementary Note 1 and Supplementary Fig. 1). As intensity in these STEM images is proportional roughly to the square of Z, the six brightest spots in the image are assigned to overlapping atomic columns of Ru and Cl atoms (Ru/Cl), whereas the three less bright spots are the Cl atomic columns (see inset in Fig. 1a). It is worth noting that for the $C2/m$ phase, reported recently in bulk $\alpha$-RuCl$_3$ crystals[21], one would expect a STEM image consisting of a well-defined line-like features with every third line substantially darker than the two others (see Supplementary Fig. 1), which is not the case in Fig. 1a. In this regard, we point out that stacking order in quasi-2D exfoliated films may not necessarily resemble the stacking order of bulk as-grown crystals, which themselves may be subject to faults that modify the layer stacking under particular crystal growth conditions. However, our neutron measurements on bulk crystals does confirm a more general possibility of $P3$-type space group scenario in $\alpha$-RuCl$_3$ system as described below.

The larger single crystals which were exfoliated for this study are consistently $C2/m$ structure at low temperatures[21,22], however, at high temperatures they comply with the $P3_1$ structure. The two structures are connected by a first-order phase transition at $T \sim 150$ K with a $\sim 30$ K hysteresis region (Fig. 2a), which explain the kinks observed in past susceptibility measurements[19]. This structural transition is characterized by the $(1,0,L = 3n)$ and $(1,1,L \neq 3n)$ peaks appearing, while $(1,1,L = 3n)$ ($n$ = integer) peaks disappearing above 150 K, as shown in Fig. 2b (see Supplementary Table 1 for the refined parameters, and methods section for details of the measurement). Indeed, the $C2/m$, $P3_112$ and $P3_1$ type space groups are polytypes[22] of each other with similar in-plane honeycomb morphology. The difference is the out-of-plane stacking arrangement. All the analysis in the following sections will be based on the $P3_1$-type space group assuming that the in-layer structural properties discussed in the text are applicable to both $C2$ and $P3$ stacks of weakly coupled $\alpha$-RuCl$_3$ layers.

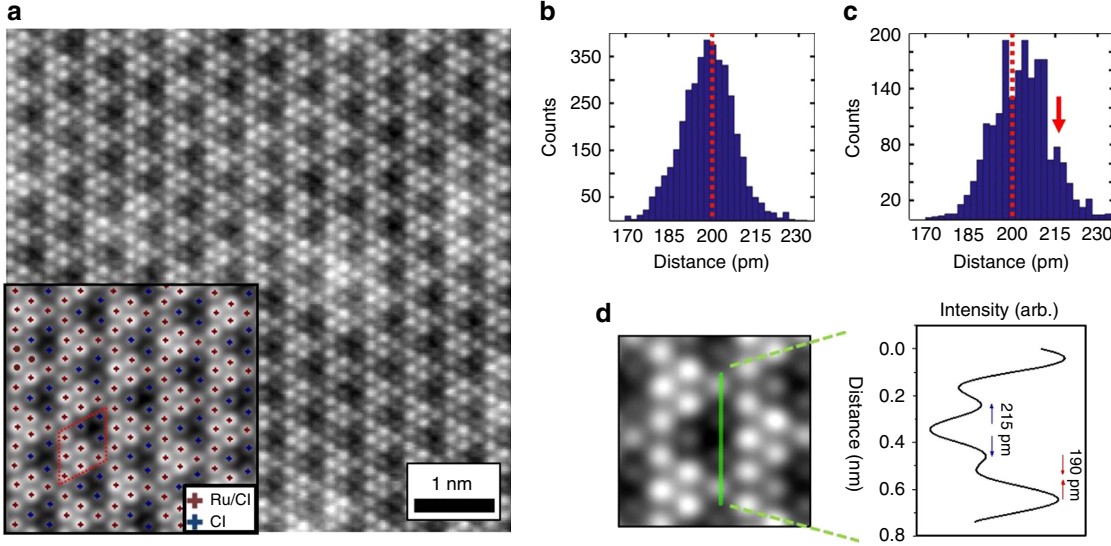

**Figure 1 | Scanning transmission electron microscopy data on α-RuCl₃ films.** (**a**) Z-contrast STEM image of α-RuCl₃ film (raw data). Inset shows filtered data, where brown and blue crosses correspond to the centers of Ru/Cl columns and Cl columns, respectively. (**b,c**) Histogram-based visualization of inter-column distances for six nearest neighbours of Ru/Cl columns (**b**) and Cl columns (**c**). Dotted red vertical lines show the value of inter-column distances expected for perfect trigonal symmetry. The red arrow in **c** indicates the presence of a well-defined shoulder originating from an increase in nearest neighbour distances of Cl columns. (**d**) Line profile illustrating a modulation of inter-column distances for Cl and Ru/Cl columns with respect to the value of 198 pm for ideal trigonal symmetry.

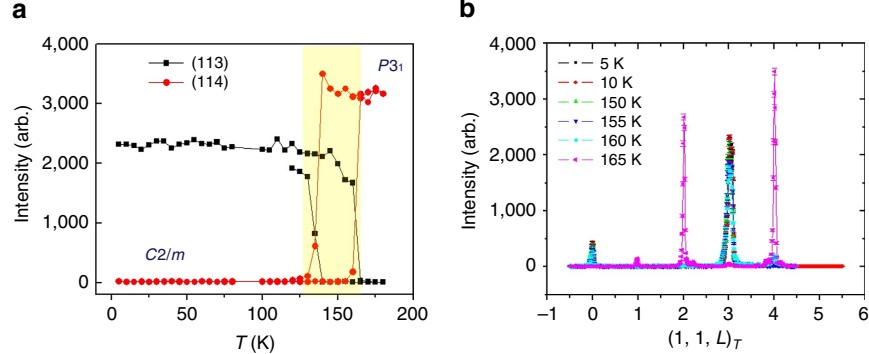

**Figure 2 | Neutron diffraction data on single crystals of α-RuCl₃.** (**a**) Temperature dependence of the (1,1,3) and (1,1,4) peak intensities. The hysteresis of the (1,1,3) and (1,1,4) orders is clearly observed (shaded area). The structural transition temperature between monoclinic and trigonal lattices is $T_s = 165$ K for warming up and $T_s = 140$ K for cooling down. (**b**) $L$-scans at several selected values of temperature.

The statistical analysis of the inter-column distances distribution performed for six nearest neighbours of Ru/Cl and Cl columns (Fig. 1b,c, respectively) shows a significant distortion from ideal trigonal symmetry (that is, the symmetry expected from a 2D-projection of atomic columns in ABC-stacked α-RuCl₃ with undistorted RuCl₆ octahedra), in which all inter-column distances are expected to be 0.198 nm. This is particularly clear from the non-symmetric nearest neighbours distance distribution for Cl columns that has a well-defined shoulder at around 215 pm (red arrow in Fig. 1c). The pairwise analysis of inter-column distances shows that the shoulder is associated with an increase in the Cl–Cl inter-column distances by $\Delta D_{\text{Cl-Cl}} \gtrsim 15$ pm. (Fig. 1d). This indicates a presence of a lattice distortion in the film system that is addressed below.

To gain further insight into the details of structural distortions found above we performed density functional theory (DFT)-based first-principles calculations on α-RuCl₃ (see Methods section for computational details) and compared them to the STEM results. Our starting point is the $P3_1 12$ unit cell with nearly perfect local cubic symmetry in which all Cl–Ru–Cl bonds in the Ru-centered

Cl octahedra are equal to $(90 \pm 1)°$ (refs 17,23). The DFT calculated relaxed structure shows that Cl atoms from opposite sublayers in each RuCl₃ 'sandwich' become displaced towards each other (Fig. 3a). This result is consistent with the findings reported in ref. 24. From additional simulations we have found that it is robust against changing the stacking and the inclusion of spin-orbit coupling, interactions and magnetism. The resultant compression of the Cl ligand cage along Ru–Ru 'bonds' ($C_3$ symmetry axis for threefold rotation) leads to a deviation from its original perfectly octahedral symmetry. Notably the lattice distortion is limited mainly to the Cl sublattice, leaving the Ru honeycomb lattice close to a distortion-free state within the resolution of available experimental structure methods. In Fig. 3c we show the DFT calculated structure of atomic columns superimposed on to the STEM experimental image of α-RuCl₃ film (top-view). One can immediately see that the direction and magnitude of a lateral component of Cl cage deformation matches well with the Cl columns displacement found from the STEM experimental image. Particularly, we found that a distance between centers of mass of Cl columns in theoretical structure is expanded

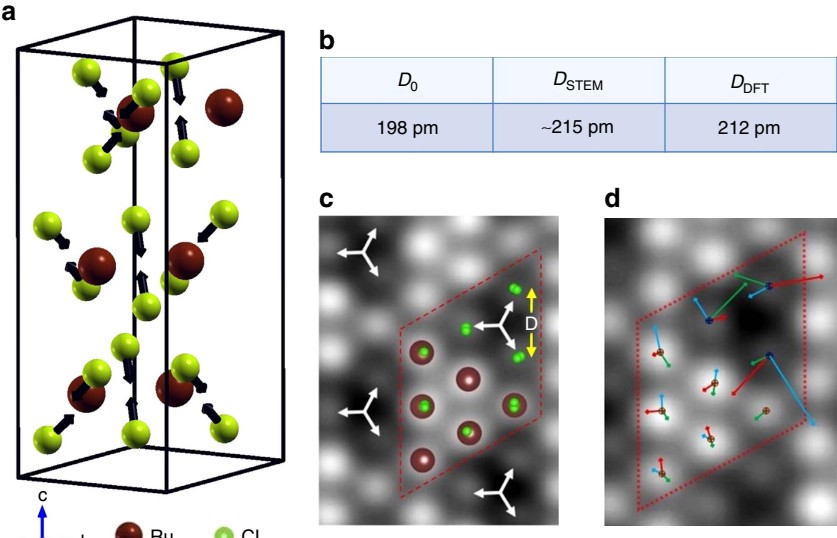

| $D_0$ | $D_{STEM}$ | $D_{DFT}$ |
|---|---|---|
| 198 pm | ~215 pm | 212 pm |

**Figure 3 | Analysis of structural distortions in α-RuCl$_3$ films. (a)** DFT-calculated relaxed structure of α-RuCl$_3$ unit cell. The equilibrium length of Ru–Cl bonds found in DFT is 236 pm. The Cl–Ru–Cl bond angle at the shared octahedral edges in the relaxed structure is ≈86.3°, whereas the rest of Cl–Ru–Cl angles are approximately 91.3°. **(b)** Cl–Cl inter-column distances in unperturbed lattice ($D_0$)[17,23], DFT calculations ($D_{DFT}$), and experimental STEM data ($D_{STEM}$). **(c)** DFT-based coordinates of ABC-stacked α-RuCl$_3$ cell superimposed on experimental STEM image. The white arrows schematically show the displacements of Cl columns. **(d)** Principal component analysis of normal modes of displacement for $M = 90$ unit cells (810 atomic columns); the center of the mass of the entire effective unit cell is chosen as origin. The principal component analysis-derived first three eigenmodes are presented as vectors of deformation from columns position in the averaged cell. Colouring scheme: first three displacement modes are described by blue, red and green arrows, respectively. The length of the arrows is magnified by a factor of 20. The dashed rhombus in **c,d** schematically denotes the effective unit cell.

by 12 pm, which is very close to an average experimental value of 15 pm (see Fig. 3b). While the presence of distortion in the bulk α-RuCl$_3$ crystals was known from earlier papers (see also Supplementary Note 2 and Supplementary Fig. 2), the STEM measurements presented here provide the first real-space evidence for octahedral distortion in thin films of α-RuCl$_3$.

The average STEM unit cell of α-RuCl$_3$ films derived above agrees well with neutron data and DFT calculations providing us a good understanding of a long range atomic order in the α-RuCl$_3$ system. We now use information contained in microscopic degrees of freedom available from STEM images to characterize atomic structure on a local scale[25–28]. Particularly, we employ a principal component analysis to search for statistically significant distortions of an average unit cell structure of STEM observations in Fig. 1a[26]. This issue is important because unit-cell scale variations in order parameters, usually not seen in spatially averaged, or reciprocal space measurements of bulk crystals, may have powerful and non-random effects on the macroscopic observables in strongly-correlated systems[29,30]. Interestingly, we found only relatively small displacements from average structure in the Ru/Cl columns (see also Supplementary Fig. 3 and Supplementary Fig. 4). As the location of Ru/Cl column centers is determined largely by a contribution from the Ru core electrons (due to the large difference in atomic numbers ($Z$) between Cl and Ru), the results in Fig. 3d suggests that the positions of Ru atoms in a honeycomb lattice remain nearly intact in a symmetric fashion. On the other hand, relatively large variations from average were observed in the positions of Cl columns. The detected local inhomogeneity in a ligand sublattice can originate from a presence of distortion nanodomains ('patches'), whose local structure, such as bond length and bond angles, deviates from the average lattice structure (see also Supplementary Note 3). A resultant interplay between local inhomogeneity and 'regular' lattice structure may lead to alternation of relative strength (and, in principle, sings) of terms in the spin-1/2 Hamiltonian. It will be interesting to see its implications for the quantum Heisenberg-Kitaev model, where a

large number of phases can be generated from the pure Kitaev limit[31] by using perturbations that are highly sensitive to the local ligand environment[32,33]. One intriguing consequence of such local perturbations is a potential realization of a spatially inhomogeneous ground magnetic state in the system. Furthermore, we suggest that our observation of the local variations in lattice structure can provide a lacking ingredient for matching theory models to the results of magnetic measurements on α-RuCl$_3$ (refs 34,35).

**Local density of states.** Having analysed in detail the local structural properties of α-RuCl$_3$ crystals, we turn to a real-space characterization of its electronic structure by virtue of ultra-high-vacuum STM imaging and spectroscopic tools. The representative STM topographic image of *in situ* cleaved α-RuCl$_3$ surface shown in Fig. 4a reveals a hexagonal lattice associated with Cl-terminated surface layer. A typical scanning tunnelling spectroscopy (STS) probe of the density of states on such surface over a wide energy region at $T = 295$ K (Fig. 4b) shows a particle-hole asymmetric d$I$/d$U$ curve with two well-defined peaks centered at about ±0.7 eV. The presence of a pronounced STS peak at ≈ −0.7 eV is in a good agreement with early angle-resolved photoemission spectroscopy (ARPES) measurements of α-RuCl$_3$, in which the author observed a quasi-flat band at 0.7 eV below the Fermi level associated with localized Ru 4$d$ states[36]. Zooming in to a narrower energy range in Fig. 4c, we found persistent peaks in LDOS at about ±0.12eV, as well as a strong suppression of density of states in the energy window of ≈0.1 eV around the Fermi level. These observations demonstrate a presence of a charge excitation gap at the Fermi level. The formation of an energy gap well above the temperature of a magnetically ordered phase indicates that α-RuCl$_3$ is a Mott-type insulator[37]. Recent DFT calculations[38] have shown that the α-RuCl$_3$ single layer remains metallic if the spin-orbit coupling and electron–electron interactions are not accounted for. The

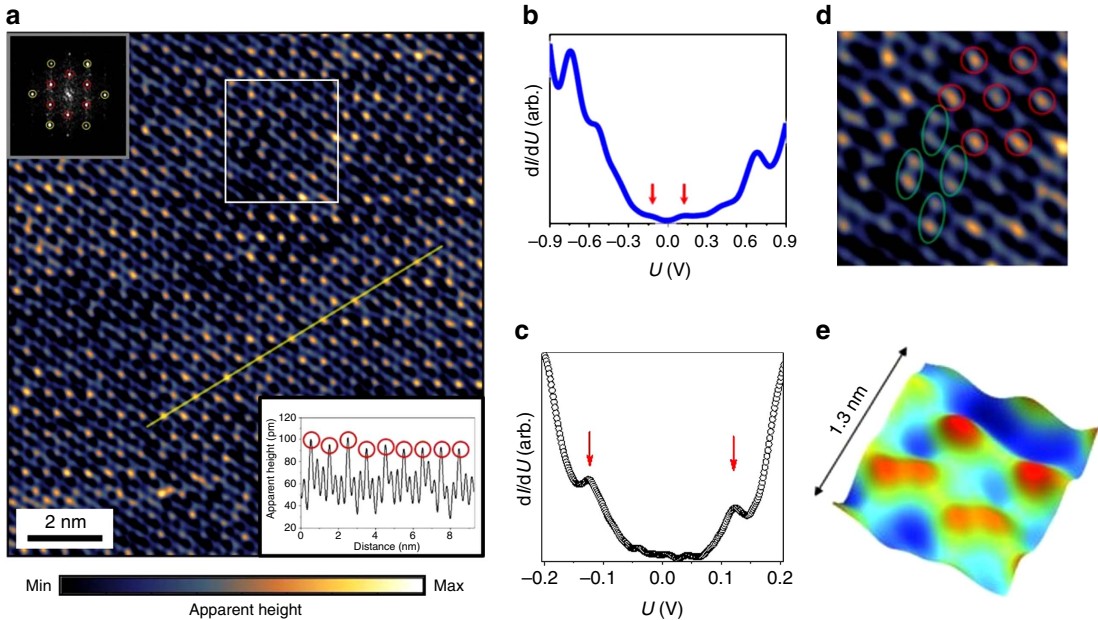

**Figure 4 | Scanning tunnelling microscopy data on *in-situ* cleaved α-RuCl₃ surface.** (**a**) Representative STM constant current image of the RuCl₃ surface layer (tunnelling conditions: $U_{tip} = 300$ mV, $I_{setpoint} = 60$ pA). Insets: 2D fast Fourier transform of the image (top left), STM cross-sectional profile along the yellow line (bottom right). (**b**) Averaged STS spectra over a wide energy window; $U = -U_{tip}$ (**c**) STS spectra in the narrower energy window around the Fermi level showing peaks at $\approx \pm 0.12$ V (denoted by red arrows). The spectra were averaged over 40 individual $dI/dU$ curves. Setpoint parameters: $U_{tip} = -500$ mV (**b**), $U_{tip} = -400$ mV (**c**); junction resistance $R_J = 25$ GΩ (**b**), 20 GΩ (**c**). (**d**) Zoomed-in area from white rectangular in **a**; coexistence of trigonal CO and dimer-like CO shown by red circles and green ovals, respectively. (**e**) Three-dimensional rendered high-resolution STM image highlighting a crossover between two different phases of CO at $U_{tip} = -1.0$ V, $I_{setpoint} = 100$ pA.

inclusion of spin-orbit coupling and electron correlations leads to an insulating character of density of states with unambiguous charge gap due to a Mott-Hubbard splitting of $J_{eff} = 1/2$ states. Therefore, our STS observations support a theory view of a monolayer α-RuCl₃ as a spin-orbit assisted Mott insulator[38]. To the best of our knowledge, this is the first observation of $J_{eff} = 1/2$ state in an individual surface layer of α-RuCl₃.

Interestingly, the $\approx 0.25$ eV surface charge gap found in our STS experiment (measured as an inter-peak separation in Fig. 4c) is much smaller than the gap of $\approx 1.2$ eV observed in recent ARPES measurements[39], which was also explained as the correlation induced Mott gap. On the other hand, the presence of $\approx 0.25$ eV gap in the STS experiment agrees well with observations of a peak at about 0.3 eV in X-ray adsorption measurements[17,40] and of a 0.2 eV gap in neutron measurements of bulk α-RuCl₃ crystals[3]. The origin of the current discrepancy between STM and ARPES measurements of the gapped state of α-RuCl₃ must be carefully addressed in future studies of this material.

**Charge density modulation**. To further explore details of the real-space electronic behaviour on a local scale it is crucial to be able to identify positions of electron density peaks associated with an underlying Cl atomic sublattice in the STM image in automated fashion. For this purpose, we first create a template unit cell from fast Fourier transform-filtered image data, and then perform normalized cross-correlation analysis coupled with position based and intensity based refinement in order to extract maximal electron densities associated with Cl sublattice (see Supplementary Note 4 for details). The resultant lattice of electronic densities is shown in Fig. 5a (see also Supplementary Note 5 and Supplementary Fig. 5 for analysis of the nearest neighbour distances in the constructed lattice).

Once the construction of electron densities lattice is completed (that is, all the positions of electron density peaks are identified), one can clearly see a modulation of STM intensity associated with an intra-unit cell symmetry breaking in a charge density distribution at three 'top' Cl atoms in the α-RuCl₃ layer unit cell. This can be confirmed by looking at the histogram-based visualization of the STM intensities distribution (Fig. 5b) for all the 'lattice' points associated with Fig. 4a. The two well-defined peaks in the histogram, whose height differs by roughly a factor of 2, reflect the enhancement of the STM intensity on every third Cl atom. On a larger scale, this leads to a formation of a charge ordered pattern (CO) with a $(\sqrt{3} \times \sqrt{3})R30\deg$ ($R3$) surface symmetry ($R$ is a translational vector of an unperturbed Cl surface lattice). Noteworthy, an emergence of a qualitatively similar $R3$ electronic superlattice has been recently reported in the STM experiment on the honeycomb iridates, where it was explained as purely due to the structural reconstructions[41]. Since STM cannot always distinguish between electronic and structural contributions, one may argue that the non-uniform distribution of STM intensity on α-RuCl₃ surface is in fact due to the geometrical tilts/rotations of Ru-centered Cl octahedral cage. However, a presence of such octahedral tilts with long range order does not follow from the detailed structural analysis described in the first part of the paper. Indeed, all Cl surface atoms remained confined to a two-dimensional plane in our structural analysis. This confirms that the formation of the $R3$ superlattice is due to the electronic symmetry breaking at Cl-terminated surface of α-RuCl₃ crystals. The superlattice and regular Cl lattice form the inner and outer hexagons, respectively, rotated by 30° with respect to each other in the 2D fast Fourier transform of the STM image (inset in Fig. 4a). The $R3$ pattern was observed up to relatively high energies with respect to Fermi level ($U_{tip} \approx |1.5 V|$), at both positive and negative bias polarities.

To elucidate a possible physical origin of the CO we start with discussing a potential relation between CO and lattice degrees of

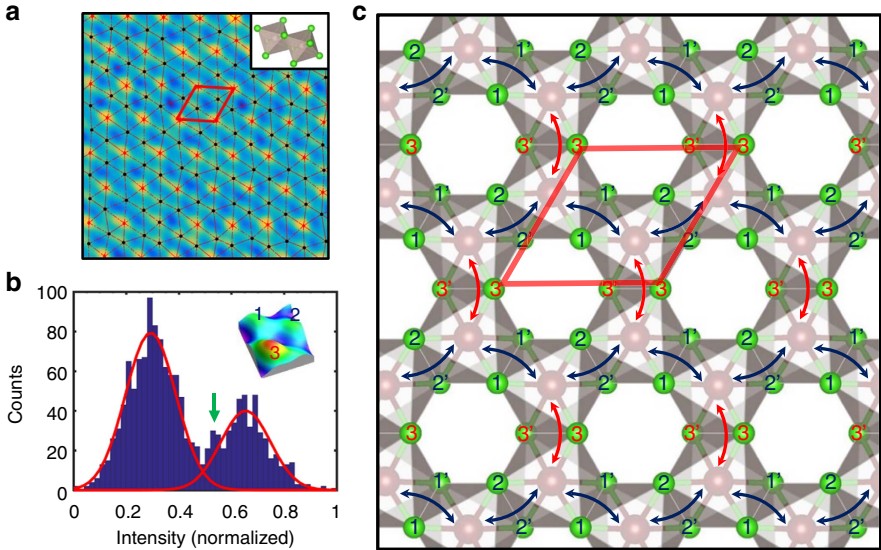

**Figure 5 | Analysis of scanning tunnelling microscopy data on α-RuCl₃ surface.** (**a**) Colour-coded binary lattice overlay. The red lattice spots correspond to the normalized intensity above the threshold of 0.5 and form a superlattice of a $(\sqrt{3}\times\sqrt{3})R30$ deg periodicity. (**b**) Histogram-based visualization of distribution of STM intensities at Cl atoms in **a**; the two peaks are approximated by Gaussian curves. The green arrow indicates admixture of a competing CO into charge density distribution. Inset in **b** shows a three-dimensional rendered experimental image of Cl triangular unit with a non-uniform distribution of STM intensities. (**c**) Schematic model showing anisotropy of charge density at Cl atoms along Ru–Cl–Ru electron hopping pathways (double-end blue and red arrows). The three 'top' Cl atoms in the unit cell are labelled in the same way in **b,c**.

freedom in α-RuCl₃. The trigonal component of RuCl₆ octahedral distortion found in DFT results in a well-defined periodic modulation of Cl interatomic distances that breaks the hexagonal symmetry of 2D surface layer. The resultant distorted hexagonal pattern features alternating short and long Cl–Cl interatomic distances of 337 pm and 359 pm, respectively (see Supplementary Note 6 and Supplementary Fig. 6). The symmetry of atomic distortions at the surface plane matches well with the symmetry of the R3 superlattice formed by the brightest spots in the STM image, implying a direct relation between Cl atomic displacements and CO. However, our DFT-based STM simulations did not reveal any imbalance in the distribution of charge density despite finding the distortion of Cl cage (see Supplementary Fig. 6), indicating that the Cl displacements are not the primary cause of the observed CO. We have also investigated the possible role of surface relaxation effects (see Supplementary Note 7, as well as Supplementary Figs 7–9) and found these to be negligible reflecting the fact that α-RuCl₃ is a quasi-2D system.

Another argument explaining the emergence of a CO phase stems from the possible non-equivalence of the three 'bond axes' (that is, the nearest neighbour Ru–Ru distances within a layer, which hereafter are referred as 'bonds') in the sample. We note that the intra-unit cell symmetry breaking in the STM experiments persists at the energies as high as +1.5 eV above the Fermi level suggesting involvement of both $t_{2g}$-derived and $e_g$-derived electronic states in a formation of the CO pattern[17]. This observation is important because a hopping process between $t_{2g}$ and $e_g$ orbital via ligand atom is vital in determining a ground state of Kitaev-Heisenberg model on the honeycomb lattice[42] (see also Supplementary Note 8).

The experimentally observed anisotropy in a charge density distribution along Ru–Cl–Ru hopping pathways, schematically denoted by red and blue arrows in Fig. 5c, indicates that the bond-dependent behaviour in α-RuCl₃ may appear at temperatures ∼3 times higher than an estimated strength of the Kitaev coupling (∼100 K) (refs 3,19). Since the STEM and neutron measurements suggest that the high-temperature lattice structure primarily complies with a trigonal P3₁ type space group and that

the Ru sublattice is isotropic (see for example, Fig. 1b), which agrees with DFT calculations, we argue that it is the anisotropic exchange pathways via the Cl sublattice that imparts additional bond-directional anisotropy along the Ru–Ru bond axes. It results in an overall anisotropic energy distribution, that could explain why the RuCl₃ layers rearrange and form a C2/m structure at temperatures below 154 K (ref. 22). Arguably, this would also determine the direction of the 'a' and 'b' axes of the C2/m structure. This, in turn, would determine the direction of the zig-zag ordering, where the blue 'bonds' in Fig. 5c have aligned spins, and the red 'bonds' have anti-aligned spins. This anisotropy in the chlorine position will directly affect the Kitaev Hamiltonian and has to be included in the calculations to simulate both the anisotropic susceptibility and the spin-wave gap of 2 meV observe in this material[3].

We next discuss a fine structure of the CO state at the Cl-terminated surface, in which the clusters of dimer-like electronic superlattice are admixed into the trigonal R3 CO electronic superlattice discussed above (see Fig. 4d,e). These two types of superlattices were persistently found in various areas of the sample at different bias voltages. The admixture of the dimer-like superlattice into CO produces an additional weaker peak in the histogram in Fig. 5b (denoted by a green arrow), as confirmed by the inspection of absolute values of the normalized STM intensities extracted from Fig. 4a. This effectively turns the surface of α-RuCl₃ into a competing CO heterogeneous system. The peculiar switching between two surface electronic super-lattices at nanometre length scales can be related to a subtle crystalline disorder, such as occurrence of vacancies in the bottom Cl layer of RuCl₃ 'sandwich' which cannot be directly seen in the STM, as well as due to the presence of defects in the second RuCl₃ 'sandwich' layer. Furthermore, the variations of an average unit cell structure in the Cl sublattice have been routinely seen in our analysis of the STEM data. The nanoscale strain fields associated with competing structural orders can induce a transition of a purely electronic nature between different CO phases as was recently reported for NbSe₂ (ref. 43). Another interesting possibility is that the observed fine structure of CO is a

reflection of magnetic short-range correlation persisting above $T_N$ (ref. 35). It remains to be seen in the future, if magnetic patches could couple via the strong spin-orbit coupling to the orbital occupation in the Ru atoms, which in turn would alter the hybridization with the Cl-$p$ orbitals and thus the tunnelling current in the STM tips. In addition, it will be educational to measure the anisotropy of the in-plane susceptibility signal, as such will also be affected by these magnetic patches. Finally, the spatial fluctuations of CO may produce an additional scattering potential for electrons in α-RuCl$_3$ and associated contribution to the resistivity of the individual α-RuCl$_3$ layers.

## Discussion

To summarize, we have presented the first comprehensive study of the local electronic and lattice degrees of freedom in α-RuCl$_3$. We expect that the findings reported here will have significant ramifications in several areas of research on spin-orbit coupled Mott insulators with possible Kitaev QSL state. First, our finding of a charge gap at $E_F$ in a surface monolayer of α-RuCl$_3$ well-above a temperature of a magnetically ordered phase supports the theoretical view of a $J_{eff} = 1/2$ Mott insulator produced by an interplay of electron correlations and spin-orbit coupling. Second, we observed the emergence of a charge-ordered pattern originating from anisotropy in the charge distribution along Ru–Cl–Ru hopping pathways. The charge order appears to have a fine structure, in which two types of the electronic superlattices, dimer-like and $R3$ trigonal, coexist at the nanometre length scale. One of the intriguing questions for future studies is: What are the exact implications of the charge distribution at the Cl- ligands on the spin degrees of freedom in the Kitaev limit? Third, our finding of a structural inhomogeneity in the ligand sublattice is significant as it may result in a unit-cell scale inhomogeneity of the magnetic ground state. Finally, our discussion on potential interplay between charge, lattice and spin degrees of freedom hinted at the possibility of nanoscale strain engineering of magnetic and electronic properties in α-RuCl$_3$, which would enable applications in hybrid spintronics and straintronics devices. In the future, we expect that extensions of this work using heterojunction techniques, as well as thin films with mis-oriented RuCl$_3$ layers[44], may provide a pathway to drive α-RuCl$_3$ or related materials towards the pure Kitaev limit where the true quantum-spin-liquid ground state flourishes.

## Methods

**Sample preparation (bulk crystals).** Commercial-RuCl$_3$ powder was purified in-house to a mixture of α-RuCl$_3$ and β-RuCl$_3$, and converted to 99.9% phase pure α-RuCl$_3$ after annealing at 500 °C. Single crystals of α-RuCl$_3$ were grown by vapour transport with TeCl$_4$ as the transport agent.

**Neutron scattering.** Neutron scattering experiments and the data in Fig. 2 were obtained using the HB-3A Four-Circle neutron diffractometer at High Flux Isotope Reactor at ORNL using an incident energy $E_i = 14.7$ meV on a large single crystal (1 cm × 1 cm × 0.3 mm) mounted on the top of the Goniometer head on a 4 K displex. The data has been analysed using FullProf structural refinement routine, and further confirmed using X-ray diffraction measurements refined using ShelX program. Above 200 K, the structure is consistent with the $P3_1$ space group, while at low temperatures the structures are consistent with $C2/m$ proposed in ref. 21. Interestingly enough, the phase transition was not observed in powder and smaller samples which remain in the same space group at all temperatures.

**STEM experiment.** The films of α-RuCl$_3$ for STEM measurements were exfoliated by sonication of bulk samples. STEM-annular dark field imaging was performed on an aberration-corrected Nion UltraSTEM-100 microscope operated at 60 kV. Only the images without noticeable electron beam induced structural damage were used for analysis. The STEM images used in analysis were summed over 30 frames after cross-correlation.

**STM experiment.** The STM experiments were performed using an Omicron VT-STM equipped with a Nanonis controller in an operating pressure of $2 \times 10^{-10}$ Torr. The samples for the STM experiments were cleaved in the ultra-high vacuum ($\sim 10^{-10}$ Torr). The STM images were obtained with mechanically cut Pt/Ir alloy tips with electrically biased tips and grounded samples. Due to the insulating nature of α-RuCl$_3$, the STS measurements were performed at $T = 295$ K, using a standard lock-in technique, with high-tunnelling junction resistance $R_J = 20$–25 Ω. To ensure a reproducibility of the high-temperature STM spectroscopic measurements, the d$I$/d$U$ curves were recorded at clean surface areas (that is, with no apparent defects) in several macroscopically different regions of each sample and the resultant averaged data was compared across the samples from three different batches.

**DFT calculations.** The DFT calculations were performed within the generalized gradient approximation using the Perdew–Burke–Ernzerhof exchange correlation scheme[45] and projector augmented wave potentials[46] as implemented in VASP[47,48]. We used a 400 eV kinetic energy cutoff and $7 \times 7 \times 3$ Monkhorst-Pack type k-point grid[49] to calculate the relaxed atomic geometries. The internal forces were relaxed to $< 10$ meV Å$^{-1}$ without enforcing any symmetry. The external lattice parameters were fixed to the experimental values reported in ref. 23.

**Statistical and image analysis.** The principal component analysis of STEM data, construction of electron densities lattice from STM image, and analysis of atomic distances were performed using MATLAB software.

**Data availability.** The data that support the findings of this study are available from the corresponding authors upon request.

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

## Acknowledgements

This research was sponsored by the Division of Materials Sciences and Engineering, Basic Energy Sciences, Department of Energy (M.Z., S.V.K., W.Z., J-Q.Y., C.B.). Research was conducted at the Center for Nanophase Materials Sciences, which also provided support (A.M., A.P.B., T.B.) and which is a DOE Office of Science User Facility. The work at ORNL High Flux Isotope Reactor was sponsored by the Scientific User Facilities Division, Office of Science, Basic Energy Sciences, U.S. Department of Energy, which supported A.B., H.C., and S.E.N. D.G.M. was supported by the Gordon and Betty Moore Foundation's EPiQS Initiative through Grant GBMF4416. This research used resources of the National Energy Research Scientific Computing Center, a DOE Office of Science User Facility supported by the Office of Science of the U.S. DOE under Contract No. DE-AC02-05CH11231. A.M. acknowledges fellowship support from the UT/ORNL Bredesen Center for Interdisciplinary Research and Graduate Education. We acknowledge Adam Aczel for the discussion during the neutron diffraction experiment and Ling Li for her help on the crystal growth. We also acknowledge useful discussions with George Jackeli and Bryan Chakoumakos.

## Author contributions

M.Z., S.V.K., and D.G.M. conceived the project. J-Q.Y., C.A.B., A.B., and D.G.M. synthesized and performed basic characterization of the bulk crystals. W.Z. performed STEM measurements on exfoliated samples. M.Z. and A.P.B. designed and performed STM experiment. A.B., H.B.C., and S.E.N. performed neutron scattering measurements. T.B. performed DFT calculations. M.Z., A.M., and S.V.K. performed statistical analysis of images. A.M. wrote the code for an automated construction of electron densities lattice from STM data. M.Z. and A.B. performed overall interpretation of all data, with guidance from S.E.N., A.P.B., and S.V.K. M.Z. and A.B. wrote the first draft of a paper, with contributions from all the authors. All authors discussed the data and its interpretation.

## Additional information

**Competing financial interests:** The authors declare no competing financial interests.

