## [Peer Review File · Nature Communications]

Reviewers' Comments:

Reviewer #1 (Remarks to the Author):

The authors performed the neutron diffraction, STEM, and STM experiments, and with the support from the first principles calculations, characterized both the long range and nanometer scale electronic structures of the a-RuCl₃ film, which turned out to have heterogeneous features consisting of two different types of charge disproportionation of Cl's.

I think this is an important piece of work on the characterization of a-RuCl₃ which is now a quite hot topic in condensed matter, as it is relevant to the Kitaev Heisenberg model hosting a quantum spin liquid in some limited parameter range.

I basically recommend this paper for publication in nature communications, while I have some questions.

- To what extent the present findings on the inhomogeneous short range charge ordered structure could be interpreted

as the picture realized in the bulk (layered) a-RuCl₃ single crystal?

The structural analysis has inconsistency regarding the transition and the symmetry above 150K, while the more microscopic analysis shows the charge inhomogeneity at higher temperatures.

I see the authors claiming that the in-plane structures may be quite similar,

but as the microscopic structure is rather peculiar/subtle (does not seem to be a regular long range order)

films have some tendency to stabilize such situation.

- The authors say that the DFT calculations after relaxation shows the ABC-stacked a-RuCl₃, but the present experiment basically deals with the thin film of 15-30nm thickness.

As the detailed information on the DFT is lacking (even in the Supplementary info), it is even hard to understand to what extent the set up of the calculations mimic the laboratory situation.

As the structure of the present material is rather fragile against possible perturbations,

I wonder whether some surface relaxation effect may strongly influence the results [c.f.].

The LDOS, in fact, seems rather regular (Fig.S5);

does the local dimer+trigonal structure possibly due to the correlation/orbital effect, may influence such results.

What should be the proper starting point only considering the spin-orbit interaction.

(Does DFT properly include spin-orbit effect?)

- Additionally, it is very useful to have an information on how the electronic structure of DFT looks like,

and how different is the localized Wannier orbitals and the overlaps between them when considering the surface

structure, since the present article aims to give a clue to the audiences to understand

how close this film system could be close to the quantum spin liquid phase in the Kitaev Heisenberg model.

Reviewer #2 (Remarks to the Author):

This is a well-rounded experimental manuscript reporting a meticulously carried out analysis of the local structural and electronic properties of alpha-RuCl₃ crystals/films using scanning transmission electron and scanning tunneling microscopies. The structural properties of alpha-RuCl₃ have attracted some interest when this material came under new scrutiny as a possible Kitaev material (such as stacking faults arising in medium-quality samples) and the further clarification of its detailed space group by local scanning probes certainly an important experimental aspect. However, the most relevant results of the manuscript at hand certainly pertain to the electronic properties. New insights in this manuscript include (i) the measurement of the local density of states and the observation of a Mott gap, which unambiguously identifies monolayer RuCl₃ as a spin-orbit entangled $j=1/2$ Mott insulator, and (ii) the direct observation of anisotropies in the charge density distribution along Ru-Cl-Ru bonds arising from an intra-unit cell symmetry breaking of the charge distribution. This latter observation is quite striking as it implies an additional level of bond anisotropy (beyond the already established spin-orbit induced anisotropic local moment couplings), which needs to be reflected in the further theoretical modeling of this material. (iii) The observation that these charge anisotropies form on temperature scales considerably higher (!) than the expected strength of the Kitaev coupling, which remains a puzzle.

With these very nice experimental findings, the manuscript should be suitable for publication in Nature Communications. In preparing a minor revision, I would like to encourage the authors to reflect on the following points:

(i) The manuscript is a tough read at the moment. Beyond some minor English problems, the paper would benefit from a stronger emphasis on the electronic structure and an early overview of the main results. Currently, the structural part dominates the first half of the manuscript, which at points reads a bit dull.

(ii) In the abstract the authors make the entirely unjustified claim that the Kitaev spin liquid phase is "known to be a promising system for topological quantum computing". This is a blunt overstatement and I challenge the authors to point to a single reference that makes a concrete proposal using these types of spin liquid Kitaev materials as a building block of a putative topological quantum computer. I think it does not help the field of spin-orbit dominated materials to come up with such exaggerated claims.

More to the point, the Majorana fermions in the gapless Kitaev spin liquid are not usable for topological quantum computations at all. For this, it needs a gapped quasiparticle that carries a Majorana zero mode (which should be carefully distinguished from a Majorana quasiparticle and even more so from the simple occurrence of a Majorana fermion operator). Such a gapped quasiparticle would be a massive vortex in the gapped spin liquid arising from the gapless Kitaev spin liquid in a properly oriented magnetic field.

(iii) While the authors correctly call for theoretical modeling which includes the experimentally observed bond anisotropies, they might also want to refer to the already existing literature on related aspects, e.g. in PRB 90, 035113 (2014).

Reviewer #3 (Remarks to the Author):

This manuscript presents the results local electronic and crystal structure of RuCl₃, which is attracting considerable attention from viewpoint of Kitaev model, using STEM and STM. The data are technically sound. The authors observed that crystal lattice at room temperature is trigonal and that a structural

phase transition to monocle structure ($C2/m$) occurs around 150 K with large hysteresis, as observed in Refs.14 and 17). They also observed local lattice distortion and charge order. These observations will contribute for understanding electronic and structural properties of RuCl_3 . However, I think that the authors do not say concrete anything how their observations are related to low-temperature magnetic properties of RuCl_3 and whether the spin-spin interaction in RuCl_3 is close to the Kitaev model. Unfortunately, at present, it is difficult to find the importance and novelty of this manuscript in quantum magnetism, which are higher than those for the papers published, for example, in Physical Review B

Reviewer #1 (Remarks to the Author):

The authors performed the neutron diffraction, STEM, and STM experiments, and with the support from the first principles calculations, characterized both the long range and nanometer scale electronic structures of the α -RuCl₃ film, which turned out to have heterogeneous features consisting of two different types of charge disproportionation of Cl's.

I think this is an important piece of work on the characterization of α -RuCl₃ which is now a quite hot topic in condensed matter, as it is relevant to the Kitaev Heisenberg model hosting a quantum spin liquid in some limited parameter range.

I basically recommend this paper for publication in nature communications, while I have some questions.

We thank the reviewer for favoring the publication of our manuscript in Nature Communications.

- To what extent the present findings on the inhomogeneous short range charge ordered structure could be interpreted as the picture realized in the bulk (layered) α -RuCl₃ single crystal? The structural analysis has inconsistency regarding the transition and the symmetry above 150K, while the more microscopic analysis shows the charge inhomogeneity at higher temperatures. I see the authors claiming that the in-plane structures may be quite similar, but as the microscopic structure is rather peculiar/subtle (does not seem to be a regular long range order) films have some tendency to stabilize such situation.

The question of whether structural, electronic and magnetic orders of bulk α -RuCl₃ material matches those found in thin films and on the surface is an important question and we thank the referee for broaching this topic. In fact, this question is universal for all STEM and STM/S measurements ranging from high-T_c superconductors to graphene and dichalcogenides. As we argue below, the results we have presented in this paper, although measured on thin films (STEM) and on the surface monolayer of α -RuCl₃ (STM/S), should be applicable to each of the quasi-isolated RuCl₃ layer inside a bulk sample. The reason for this is threefold:

(a) Based on the current experimental¹ and theoretical² (see also below our response regarding DFT calculations) understanding, it is reasonable to assume that the α -RuCl₃ can be treated as a quasi-2D material, in which each layer is nearly independent of the others. Taking into account the absence of dangling bonds, as well as the absence of alkali atomic layers above and below each RuCl₃ layer (in sharp contrast with iridates), we do not have any reason to believe that such quasi-2D system should develop any additional disorder *specific* to exfoliation/cleavage and reduction of the number of layers. This line of thinking is further supported by reports on similar

layered compounds TiCl_3 and VCl_3 ,³ as well as by conclusions of Weber *et al.*⁴ that an in-plane structure of $\alpha\text{-RuCl}_3$ is maintained in the exfoliation process. Thus the modulations of structural and electronic orders reported in our study are likely to be found in each of quasi-2D layers that comprise a bulk sample.

(b) In regards to the structural transition at 150 K, it occurs because of slight re-organization of the planes. Since the change is mostly from a change in stacking formation the energy involved is rather small, < 1 meV/unit-cell.² This is further supported by inverse susceptibility data (for example, see Fig. S1 in study of Banerjee *et al.*⁵, or Fig. 2 in a paper of Kubota *et al.*⁶) which shows that, functionally, the form and the anisotropy of the magnetic susceptibility remain unchanged across the transition. The magnetic susceptibility, and the anisotropy, arises almost completely from the in-plane interactions (for example, see discussion in a paper by Weber *et al.*⁴), and any drastic change in the local environment from the phase transition should have also affected the overall behavior of the susceptibility through the 150 K transition. Since that is not the case, it provides further support for the fact that the 150 K transition is inconsequential for the overall structural and magnetic analysis of quasi-2D layers of $\alpha\text{-RuCl}_3$.

c) Finally, please note here, that the inhomogeneity in electronic charge order we report is statistically significant and is systematically present throughout the sample. This is not an artifact of a specific region of a sample which would have pointed to cleavage resulted damage. A surface-driven argument can be further negated based on new DFT calculations which we will present below. Thus, as far as the local inhomogeneity is concerned, we discuss our results within quite a commonly accepted framework in which local microscopic probes (such as STEM and STM) can detect minute variations from average structure (symmetry) that may simply not be seen (averaged out) in the spectroscopic, or reciprocal space, measurements of bulk samples.⁷

For all the above points our study of structural/electronic orders in thin films and on the surface at room temperature is likely to be relevant for the bulk and for a wide range of temperatures. While our study unambiguously establishes the presence of peculiar nanoscale (dis-)order, and discusses its potential implications to underlying magnetic Hamiltonian, this definitely opens the room for a more detailed investigation regarding the exact forms of this (dis-)order which would be a subject of future studies.

We have added a part (a) of this discussion in the supplementary section S.8 to clarify this issue.

- The authors say that the DFT calculations after relaxation shows the ABC-stacked α -RuCl₃, but the present experiment basically deals with the thin film of 15-30nm thickness.

As mentioned in the point (b) of the previous section, and elucidated in the paper by Kee and Kim using DFT², the out-of-plane interactions are rather weak, and the difference between the two unit cells from the perspective of in-plane structure should be rather small. Given this pre-information, in the present study, we did not investigate with DFT whether the ABC stacking is energetically more favorable than another type of stacking. Instead we use the experimentally observed at room temperature (see Fig. 1a, Fig. S1, and also Fig. 2) $P3_112$ unit cell (with ABC stacking) as a starting point to investigate the atomic distortions within the RuCl₃ layers. However, we did attempt to perform DFT with a $C2/m$ starting point as well. Not surprisingly, our conclusion has always been that **the contraction of Cl atoms illustrated in Fig 3a in the manuscript are extremely robust against factors such as stacking order ($C2/m$ vs. $P3_112$), relaxation of lattice parameters, spin-orbit coupling and interactions.** To clarify this point to the reader, we have added one line in the manuscript to reinforce this point in Page #7 (line # 201).

We do agree that a DFT based investigation of the stacking, which simultaneously encompasses the modulations, on the other hand is an important problem that we did not address and we certainly cannot preclude a possibility that it could be sensitive to factors such as the film thickness. We leave these subtle issues to future investigations and focus our DFT studies in this paper only on the matter of the intra-layer (i.e. in-plane) atomic displacements, as these are the type of distortions that we are inferring from our microscopic experiments.

As the detailed information on the DFT is lacking (even in the Supplementary info), it is even hard to understand to what extent the set up of the calculations mimic the laboratory situation.

The details are given in the Methods section. To better clarify this we now refer to the Methods section in the part of the text that discusses the DFT results in Page #7 (line #196).

As the structure of the present material is rather fragile against possible perturbations, I wonder whether some surface relaxation effect may strongly influence the results [c.f.]. The LDOS, in fact, seems rather regular (Fig.S5); does the local dimer+trigonal structure possibly due to the correlation/orbital effect, may influence such results. What should be the proper starting point only considering the spin-orbit interaction. (Does DFT properly include spin-orbit effect?)

To demonstrate theoretically the insensitivity of the LDOS with respect to surface relaxations we added an extra section in the supplement (see section S.7) in which we compare the spatially and orbitally resolved DOS from a bulk and a slab of α -RuCl₃. The LDOS from the slab and the bulk calculation are very similar and neither of them shows a reordering of the charge. **Therefore surface relaxations do not seem to be a possible explanation of the experimentally observed charge order.** Instead we think a proper starting point in terms of spin-orbit coupling and interactions is a more urgent thing to look at, as also suggested by the

referee. As we speculate in the end of the manuscript, the observed dimer+trigonal charge order(s) could be associated with a magnetic short-range correlations persisting above T_N . Via a strong spin-orbit coupling the magnetic structure could alter the orbital occupation of the Ru atoms and thus explain the experimentally observed charge structures. Unfortunately, in the currently used DFT code (VASP) it is not possible to extract the LDOS (= simulated STM image) in the presence of spin-orbit coupling. In future investigations we intend to find a work-around for this technical problem and compute the LDOS in the presence of interactions and spin-orbit coupling. In the mean time we believe that the presentation to the community of the important experimental findings in this study should not be delayed because of that.

- Additionally, it is very useful to have an information on how the electronic structure of DFT looks like, and how different is the localized Wannier orbitals and the overlaps between them when considering the surface structure, since the present article aims to give a clue to the audiences to understand how close this film system could be close to the quantum spin liquid phase in the Kitaev Heisenberg model.

We thank the referee for this comment and note that there have been several reports on the electronic structure already. In particular Ref. (1) in the manuscript gives a detailed account on the influence of interactions and spin-orbit coupling on the electronic structure. In section S.7 that was added to our revised supplement we compare the DOS from the Ru-d orbitals in bulk α - RuCl_3 and from the Ru-d orbitals in the surface layer of the three RuCl_3 layer slab with 10 Ang of vacuum. Specifically, we note that the Ru- t_{2g} band width in both cases are very similar (roughly within [-1.0,0.2] eV) indicating that the overlaps between the Ru- t_{2g} Wannier orbitals in the bulk are very similar to those in the surface again reflecting the quasi 2D-ness of α - RuCl_3 .

Reviewer #2 (Remarks to the Author):

This is a well-rounded experimental manuscript reporting a meticulously carried out analysis of the local structural and electronic properties of alpha-RuCl3 crystals/films using scanning transmission electron and scanning tunneling microscopies. The structural properties of alpha-RuCl3 have attracted some interest when this material came under new scrutiny as a possible Kitaev material (such as stacking faults arising in medium-quality samples) and the further clarification of its detailed space group by local scanning probes certainly an important experimental aspect. However, the most relevant results of the manuscript at hand certainly pertain to the electronic properties. New insights in this manuscript include (i) the measurement of the local density of states and the observation of a Mott gap, which unambiguously identifies monolayer RuCl3 as a spin-orbit entangled $j=1/2$ Mott insulator, and (ii) the direct observation of anisotropies in the charge density distribution along Ru-Cl-Ru bonds arising from an intra-unit cell symmetry breaking of the charge distribution. This latter observation is quite striking as it implies an additional level of bond anisotropy (beyond the already established spin-orbit

induced anisotropic local moment couplings), which needs to be reflected in the further theoretical modeling of this material. (iii) The observation that these charge anisotropies form on temperature scales considerably higher (!) than the expected strength of the Kitaev coupling, which remains a puzzle.

With these very nice experimental findings, the manuscript should be suitable for publication in Nature Communications. In preparing a minor revision, I would like to encourage the authors to reflect on the following points:

We thank the reviewer for a careful scrutiny and providing above this very nice summary of our work. We are grateful for the highly positive comments on our work and for recommending publication of our manuscript in Nature Communications.

(i) The manuscript is a tough read at the moment. Beyond some minor English problems, the paper would benefit from a stronger emphasis on the electronic structure and an early overview of the main results. Currently, the structural part dominates the first half of the manuscript, which at points reads a bit dull.

We highly appreciate the suggestion of the referee to improve the readability of the manuscript. To address this issue we have significantly shortened the section on the structural measurements and the technical details relevant to the measurements in the beginning of the manuscript. We have systematically moved some of these details to the ‘methods’ section and the figure captions. Instead, to emphasize the electronic results right in the beginning, we have added an early overview of the results, as suggested by the reviewer, in the last paragraph of introduction (Page #5)

With these changes, we think that the manuscript will lead the readers more directly to the interesting results.

(ii) In the abstract the authors make the entirely unjustified claim that the Kitaev spin liquid phase is "known to be a promising system for topological quantum computing". This is a blunt overstatement and I challenge the authors to point to a single reference that makes a concrete proposal using these types of spin liquid Kitaev materials as a building block of a putative topological quantum computer. I think it does not help the field of spin-orbit dominated materials to come up with such exaggerated claims.

More to the point, the Majorana fermions in the gapless Kitaev spin liquid are not usable for topological quantum computations at all. For this, it needs a gapped quasiparticle that carries a Majorana zero mode (which should be carefully distinguished from a Majorana quasiparticle and even more so from the simple occurrence of a Majorana fermion operator). Such a gapped

quasiparticle would be a massive vortex in the gapped spin liquid arising from the gapless Kitaev spin liquid in a properly oriented magnetic field.

We take the referee's point on board. The phrase that the Kitaev spin liquid phase is "known to be a promising system for topological quantum computing" has been removed from the abstract, and has been replaced with the phrase "...is proposed to host the celebrated 2D Kitaev model which has an elusive quantum spin liquid ground state, and fascinating physics relevant to the development of future templates towards topological quantum bits."

We do note that the huge interest in topological quantum computing that has come from various strands of Kitaev's work is a testament to the importance of this field, and it needs to be mentioned to pay the correct tribute. Additionally, there are recent theoretical attempts (such as strain-induced gapping of the Kitaev QSL excitations⁸) to extend the relevance of this model beyond just the academic interest towards possible quantum information storage. Our study of thin films and surface properties will enable such studies in the future. We referred to some of these researches in the introduction in the revised manuscript when mentioning about the potential usefulness of our results towards this interesting problem.

(iii) While the authors correctly call for theoretical modeling which includes the experimentally observed bond anisotropies, they might also want to refer to the already existing literature on related aspects, e.g. in PRB 90, 035113 (2014).

We thank the referee for drawing our attention towards this very interesting paper. This PRB study discusses a sensitivity of the collective spin-orbital states of the honeycomb Kitaev-Heisenberg model to small perturbations in lattice structure and is therefore directly relevant to findings of our manuscript. We have referred to this study in the revised version of our manuscript.

Reviewer #3 (Remarks to the Author):

This manuscript presents the results local electronic and crystal structure of RuCl₃, which is attracting considerable attention from viewpoint of Kitaev model, using STEM and STM. The data are technically sound.

We thank the referee for praising the technical/experimental rigour of our study.

The authors observed that crystal lattice at room temperature is trigonal and that a structural phase transition to monocle structure (C2/m) occurs around 150 K with large hysteresis, as observed in Refs.14 and 17). They also observed local lattice distortion and charge order. These observations will contribute for understanding electronic and structural properties of RuCl₃.

However, I think that the authors do not say concrete anything how their observations are related to low-temperature magnetic properties of RuCl₃ and whether the spin-spin interaction in RuCl₃ is close to the Kitaev model. Unfortunately, at present, it is difficult to find the importance and novelty of this manuscript in quantum magnetism, which are higher than those for the papers published, for example, in Physical Review B

We report two novel discoveries on α -RuCl₃, arguably one of the most interesting proposed materials for realizing the Kitaev spin liquid:

1) We observed substantial deviations of the Ru-Cl bonds away from perfect local cubic symmetry in thin films of α -RuCl₃.

(2) We observed an unexpected charge order in $J_{\text{eff}}=1/2$ Mott phase in a single RuCl₃ surface layer.

Undoubtedly, these findings will contribute to a better understanding of this material as also the Referee acknowledges. Most importantly, a complete physical model of α -RuCl₃ must now take these observations into account. **Particularly, the charge anisotropy which we discovered will be a new input into future calculations that will try to understand the nature of quantum magnetism in this material.** Therefore, we believe our manuscript will have a large impact. We offered our current understanding on the connection of our observations to low T quantum magnetism in the supplementary section S.8. Given the fast pace of this field, we believe our findings are urgent and will encourage a series of intermediate research that will finally lead to the proper understanding of this important problem.

References:

- (1) Banerjee, A.; Bridges, C. A.; Yan, J. Q.; Aczel, A. A.; Li, L.; Stone, M. B.; Granroth, G. E.; Lumsden, M. D.; Yiu, Y.; Knolle, J.; Bhattacharjee, S.; Kovrizhin, D. L.; Moessner, R.; Tennant, D. A.; Mandrus, D. G.; Nagler, S. E. Proximate Kitaev quantum spin liquid behaviour in a honeycomb magnet. *Nat Mater* **2016**, *15*, 733-740.
- (2) Kim, H.-S.; Kee, H.-Y. Crystal structure and magnetism in α -RuCl₃: An *ab-initio* study. *Physical Review B* **2016**, *93*, 155143.
- (3) Zhou, Y.; Lu, H.; Zu, X.; Gao, F. Evidencing the existence of exciting half-metallicity in two-dimensional TiCl₃ and VCl₃ sheets. *Scientific Reports* **2016**, *6*, 19407.
- (4) Weber, D.; Schoop, L. M.; Duppel, V.; Lippmann, J. M.; Nuss, J.; Lotsch, B. V. Magnetic Properties of Restacked 2D Spin 1/2 honeycomb RuCl₃ Nanosheets. *Nano Letters* **2016**, *16*, 3578-3584.
- (5) Banerjee, A.; Yan, J.; Knolle, J.; Bridges, C. A.; Stone, M. B.; Lumsden, M. D.; Mandrus, D. G.; Tennant, D. A.; Moessner, R.; Nagler, S. E. Neutron tomography of magnetic Majorana fermions in a proximate quantum spin liquid. *ArXiv e-prints* 2016.
<http://adsabs.harvard.edu/abs/2016arXiv160900103B>.

- (6) Kubota, Y.; Tanaka, H.; Ono, T.; Narumi, Y.; Kindo, K. Successive magnetic phase transitions in α -RuCl₃: XY-like frustrated magnet on the honeycomb lattice. *Physical Review B* **2015**, *91*, 094422.
- (7) Belianinov, A.; He, Q.; Kravchenko, M.; Jesse, S.; Borisevich, A.; Kalinin, S. V. Identification of phases, symmetries and defects through local crystallography. *Nat Commun* **2015**, *6*.
- (8) Rachel, S.; Fritz, L.; Vojta, M. Landau Levels of Majorana Fermions in a Spin Liquid. *Physical Review Letters* **2016**, *116*, 167201.

Reviewers' Comments:

Reviewer #1 (Remarks to the Author):

The authors have answered to all questions addressed, and the presentation of the manuscript is improved. I think that the manuscript is ready for publication.